# Structure and Assembly of the *Proteus mirabilis* Flagellar Motor by Cryo-Electron Tomography

**DOI:** 10.3390/ijms24098292

**Published:** 2023-05-05

**Authors:** Mohammed Kaplan, Qing Yao, Grant J. Jensen

**Affiliations:** 1Division of Biology and Biological Engineering, California Institute of Technology, Pasadena, CA 91125, USA; 2Department of Chemistry and Biochemistry, Brigham Young University, Provo, UT 84604, USA

**Keywords:** bacterial flagellar motor, cryo-electron tomography, *Proteus mirabilis*

## Abstract

*Proteus mirabilis* is a Gram-negative Gammaproteobacterium and a major causative agent of urinary tract infections in humans. It is characterized by its ability to switch between swimming motility in liquid media and swarming on solid surfaces. Here, we used cryo-electron tomography and subtomogram averaging to reveal the structure of the flagellar motor of *P. mirabilis* at nanometer resolution in intact cells. We found that *P. mirabilis* has a motor that is structurally similar to those of *Escherichia coli* and *Salmonella enterica*, lacking the periplasmic elaborations that characterize other more specialized gammaproteobacterial motors. In addition, no density corresponding to stators was present in the subtomogram average suggesting that the stators are dynamic. Finally, several assembly intermediates of the motor were seen that support the inside-out assembly pathway.

## 1. Introduction

*Proteus mirabilis* is a dimorphic Enterobacteriaceae that can differentiate from a small rod shape capable of swimming in liquid environments into a larger hyperflagellated cell that swarms on solid surfaces [1]. In fact, the name of this bacterium is derived from Proteus, who was a sea god in Greek mythology known for his ability to change his shape to avoid being captured. *P. mirabilis* is a major causative agent of various urinary tract infections such as pyelonephritis and catheter-associated urinary tract infections (CAUTI) which represent about 40% of hospital-acquired infections [2,3,4]. Many features contribute to the ability of *P. mirabilis* to cause CAUTI, including their capacity to produce fimbriae which allow them to adhere to surfaces (the *P. mirabilis* genome encodes 17 fimbrial operons [5]), the production of the enzyme urease which helps in forming crystalline biofilms that block catheters [6,7], and their flagellar-dependent swarming motility on solid surfaces which enables them to move across catheters to the urinary tract [8,9].

Both the swimming and swarming motilities of *P. mirabilis* are flagellar-dependent. The bacterial flagellum is one of the major motility nanomachines in bacteria and consists of three structurally-distinct parts: a cell-envelope-embedded motor that generates torque to rotate a long extracellular filament, connected by a flexible joint known as the hook [10]. The motor comprises two further parts: a rotor and the stators. The rotor consists of a series of rings including the cytoplasmic (C-ring) and the inner membrane embedded ring (the membrane/supramembrane ring, also known as the MS-ring). Extending from the MS-ring in the periplasm is a driveshaft known as the rod, which is surrounded by two other rings that act as bushings during the motor rotation and form a relic upon flagellar disassembly [11,12,13,14], the P- (peptidoglycan) and L- (lipopolysaccharide) rings. Once the L-ring assembles it makes a hole in the outer membrane through which the hook and filament subunits are secreted to assemble outside the cell [15]. Embedded inside the MS-ring is a dedicated type 3 secretion system which secretes the rod, hook, and filament proteins through the inner membrane. In addition, many other chaperone, junction and capping proteins are involved in flagellar biogenesis [10,16,17]. The stators are ion channels that are embedded in the inner membrane that rotate while pumping ions (such as H^+^ or Na^+^) across the inner membrane and that interact with the upper part of the C-ring in the cytoplasm, which results in torque generation and, ultimately, filament rotation [18,19,20,21].

While all known flagellar motors share a conserved core, which is almost equivalent to the motors of *E. coli* and *Salmonella*, different species incorporate various periplasmic or extracellular elaborations to allow the generation of the optimal amount of torque required to move the cell efficiently in its environment [22,23,24,25]. For example, pathogens such as *Helicobacter pylori*, *Campylobacter jejuni*, *Legionella pneumophila*, and various *Vibrio* spp., assemble extra periplasmic rings that stabilize a wide stator ring allowing the generation of high torque [22,23,26,27]. On the other hand, the flagellar motors of *E. coli* and *Salmonella* have dynamic stators that use a catch bond mechanism to sense the viscosity of the external milieu and engage around the motor in a number proportional to that viscosity [28,29].

Despite the importance of the *P. mirabilis* flagellar motor in its swimming and swarming modes of motility, and thereby in its pathogenicity, the structure of this motor has not yet been characterized. Here, we used cryo-electron tomography (cryo-ET), which enables the direct visualization of macromolecular complexes in cellular settings to nanometer resolution in a frozen-hydrated state [30,31], to image the flagellar motor of swarming *P. mirabilis* cells. We found that this motor is structurally akin to those present in *E. coli* and *Salmonella* in that it only has the conserved core and lacks the periplasmic and extracellular elaborations described in other species. Furthermore, the subtomogram average of the motor is devoid of densities that correspond to stators, suggesting that they are dynamic in the inner membrane, such as those of the *E. coli* motor. Similarly to what has been recently suggested for *E. coli* and *Salmonella* [32], we hypothesize that the presence of this general-purpose motor with dynamic stators benefits *P. mirabilis* and enables it to colonize and move in diverse environments.

## 2. Results and Discussion

To visualize the flagellar motor of *P. mirabilis* cells, we imaged them in a frozen-hydrated state using cryo-ET. Initially, we visualized planktonic swimming cells. These cells were ~800–1000 nm long and were characterized by multiple (hundreds) of thin peritrichous proteinaceous extensions that were a few hundred nanometers long and ~5 nm wide (Appendix A). We could not identify anything in the periplasm at the base of these extensions. Presumably, these are the fimbriae that are known to be associated with vegetative cells, as *P. mirabilis* has 17 fimbriae operons encoded in its genome [5]. On the other hand, flagella were very sparse in these cells. This is in accordance with previous studies that indicated the reciprocal expression of flagellar and fimbriae genes in *P. mirabilis* [33,34,35,36].

The absence of flagella in planktonic cells excluded the possibility of performing subtomogram averaging to obtain an enhanced signal-to-noise structure of the motor. For this reason, we resorted to imaging swarmer *P. mirabilis* cells which are known to be hyperflagellated [1]. Compared to planktonic cells, swarmer cells were larger in size (>1000 nm long) and exhibited tens of flagella with no fimbriae (Figure 1A and Appendix A). The presence of high numbers of flagella allowed us to average 41 motors at the base of the flagellar filaments and obtain an average structure with a higher signal-to-noise ratio compared to individual particles, revealing the macromolecular architecture of the motor (Figure 1B). The motor has an architecture similar (at the low resolution of our cryo-tomograms and subtomogram average) to those of *E. coli* and *Salmonella* where only the conserved core is present without any periplasmic or extracellular embellishments (Figure 1B). Moreover, no clear stator density could be resolved in our low-resolution subtomogram average (Appendix A), suggesting that they could be dynamic with variable numbers being recruited around the motor at any given moment, such as in *E. coli*.

In addition, the presence of multiple flagella in swarmer cells allowed us to capture motors at different assembly stages. We identified a subcomplex containing the C-and MS-rings (the C-complex, [37]), a subcomplex containing the C-complex with the rod and the P-ring (the P-complex, [37]), the basal body where a hole in the outer membrane is visible, and the hook-basal body (Figure 1C–G). These stages are in accordance with the inside-out assembly pathway of the flagellar motor described in many other species including *E. coli* and *Salmonella* [37,38,39].

It has recently been indicated that the motors of *E. coli* and *Salmonella* are not native to these species but were acquired from Betaproteobacteria via lateral gene transfer [32]. Hence, we compared our structure of the *P. mirabilis* motor to the motors of *E. coli*, *Salmonella*, and the recently published structure of the Betaproteobacterium *Bordetella bronchiseptica* motor [32]. While all the motors look similar at this resolution, one notable difference between the motors of *P. mirabilis* and *B. bronchiseptica* on one hand, and those of *E. coli* and *Salmonella* on the other, as judged at the resolution of the available subtomogram averages, was in the shape of the C-ring (Figure 2). While the C-ring of the latter two motors appeared straight with a constant diameter of ~40 nm, the lower part of the C-ring of both *P. mirabilis* and *B. bronchiseptica* appeared wider (~42 nm) compared to the upper part of the C-ring (~38 nm, Figure 2).

The C-ring of the motor of *E. coli* consists of three proteins: FliG (located close to the inner membrane), FliM (in the middle of the C-ring), and FliN (at the base of the C-ring) [40]. Interestingly, both FliN and FliM continuously exchange between the C-ring and a cytoplasmic pool [41,42,43]. It will be interesting to see if FliM and FliN also turn over in *P. mirabilis* and if this is related to the wider bottom of the C-ring, presumably also formed by FliN, in swarmer *P. mirabilis*. Presumably, increasing the stoichiometry of FliM can make the C-ring more sensitive to the active Che-Y protein, which binds to FliM, promoting the motor to switch spinning direction [42].

The general-purpose motor of Enterobacteriaceae, believed to have been acquired from Betaproteobacteria via lateral gene transfer, is thought to have enabled Enterobacteriaceae to inhabit and thrive in a wide range of environments [32]. Perhaps having a general-purpose motor helps *P. mirabilis* to switch from swimming motility in liquid media to swarming on solid surfaces (such as catheters), where each environment requires different torque, and thereby different numbers of stators engaged around the motor. A highly specialized motor, with a fixed stator ring, might be a disadvantage when the microbe encounters such diverse environmental conditions.

Finally, it would be interesting to obtain the structures of the flagellar motors of both the vegetative and swarmer cell types in the future at higher resolution (compared to what we report here) to see if there are any structural changes that occur in the motor between the two states.

## 3. Materials and Methods

### 3.1. Strains and Growth Conditions

*Proteus mirabilis* strain HI4320 was used in this study. Vegetative swimming cells were grown in Luria Broth (LB) medium at 37 °C with continuous shaking at 110 rpm to OD_600_ of 1.0. Swarmer cells were prepared as described previously in [44]. Briefly, we inoculated 4 μL of a suspension of ∼4 × 10^5^
*P. mirabilis* cells/ml onto the surface of a 1.5% (wt/vol) Bacto Difco agar gel containing nutrient broth. The agar gel was prepared using 50 ml of hot swarm agar which was pipetted into 150 by 15 mm petri dishes (Becton, Dickinson), and left to solidify at 25 °C for 30 min. The plates were incubated at 30 °C with 90% relative humidity in an incubator (without shaking) after the absorption of the inoculum into the agar.

Swarm cells were harvested from the smooth leading edge of a migrating colony on the agar plate after 15 h. We used a 1 μL calibrated inoculation loop (220215; Becton, Dickinson, Franklin Lakes, NJ, USA) to harvest the cells. Subsequently, the cells were removed from the loop by rinsing with 500 μL of motility buffer (0.01 M KPO_4_, 0.067 M NaCl, 10^−4^ M EDTA, pH 7.0, containing 0.1 M glucose and 0.001% Brij-35) and centrifuged for 10 min at 1500× *g*. While this established protocol has been used to study the motility of swarmer cells, we cannot exclude the possibility that any of these treatments has any effect on the structure of flagellar motor.

### 3.2. Cryo-ET Sample Preparation and Imaging

Resuspended cells were mixed with 10 nm gold beads coated with bovine serum albumin and ~4 μL of this mixture was applied on a glow-discharged 200 mesh Quantifoil grid and plunge-frozen in a mixture of liquid ethane and propane [45]. Imaging was performed using a 300 keV Polara transmission electron microscope (FEI) equipped with a GIF energy filter (Gatan) and a K2 Summit direct detector (Gatan). Data collection was performed using UCSF Tomography software [46] and tilt series were collected from −60° to +60° in 1° increments with a defocus of −8 μm and total dosage of 160 e^−^/Å^2^. The reconstruction of tomograms was carried out using IMOD software [47], and subtomogram averaging with twofold symmetrization along the particle y-axis was performed using PEET program [48].

## Figures and Tables

**Figure 1 ijms-24-08292-f001:**
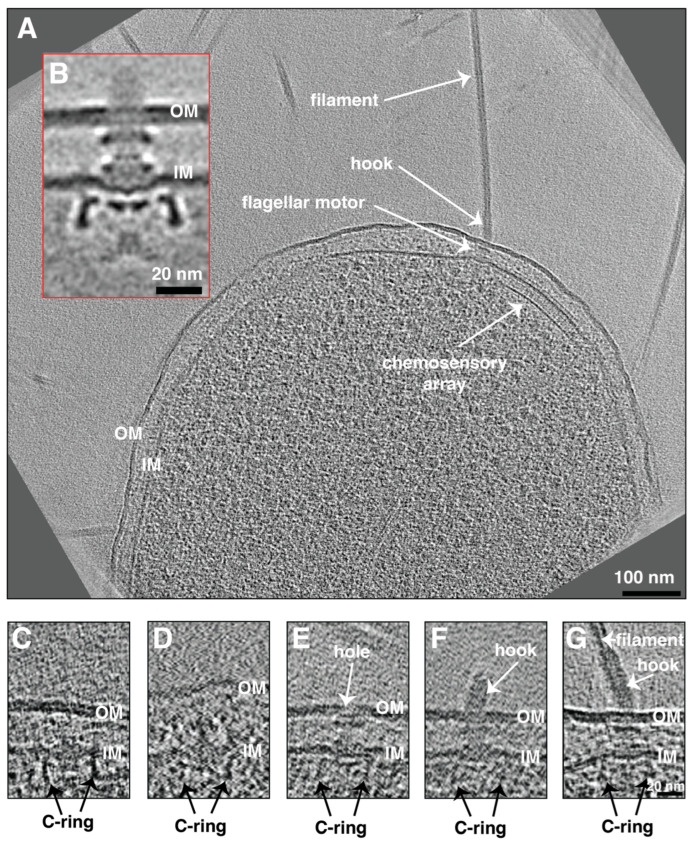
(**A**) A slice through a cryo-electron tomogram of a *P. mirabilis* swarmer cell highlighting the presence of multiple flagella and chemosensory arrays. Scale bar is 100 nm. (**B**) A central slice through the subtomogram average of swarmer *P. mirabilis* flagellar motor. Scale bar is 20 nm. (**C**–**G**) Slices through cryo-electron tomograms of *P. mirabilis* cells illustrating the presence of different assembly stages starting from the C-complex (**C**), the P-complex (**D**), basal body (**E**), hook-basal body (**F**), and fully assembled motor (**G**). Scale bar is 20 nm (valid through (**C**–**G**)). OM = outer membrane, IM = inner membrane.

**Figure 2 ijms-24-08292-f002:**
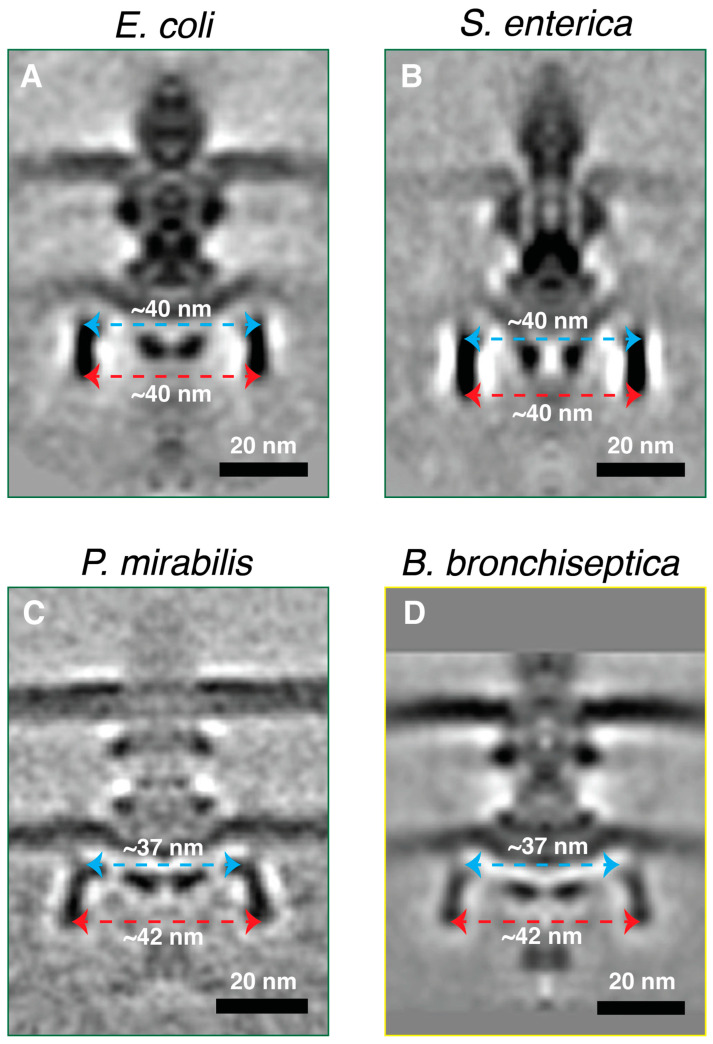
(**A**–**D**) Central slices through subtomogram averages of the flagellar motors of *E. coli* (EMD-5311), *Salmonella* (EMD-5310), *P. mirabilis*, and *B. bronchiseptica* (EMD-4999), respectively. A green outline indicates a Gammaproteobacterial motor; a yellow outline indicates a Betaproteobacterial motor. The diameters of the lower part (red arrows) and the upper part (light blue arrows) of the C-ring are indicated. Scale bar is 20 nm.

## Data Availability

All tomograms are available upon request.

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
