# Peer review of "Structure and Assembly of the Proteus mirabilis Flagellar Motor by Cryo-Electron Tomography"

_ijms, 2023, doi:10.3390/ijms24098292_

Round 1
Reviewer 1 Report
The authors present the in situ structure of the Proteus mirabilis flagellar motor that was obtained by cryo-electron tomography and subtomogram averaging. Like the flagellar motors of Escherichia coli and Salmonella enterica, the P. mirabilis motor contains only the core structures and lacks the motor accessories found in many other bacterial species. The new information in the manuscript provides a minimal contribution to our understanding of the bacterial flagellum, but to my knowledge is the first in situ structure of the P. mirabilis flagellar motor and therefore has merit.
Specific comments:
1. The authors indicate that the in situ structure of the P. mirabilis motor lacks stators. Given that the images of the motor were obtained from swarmer cells (i.e., condition where there is a high load on the motor), one would expect there to be numerous stator units associated with the motor. It is not clear from the images in Figures 3B and 5C that the stators are not present. In both images, there is a density linking the top of the C-ring to the outer membrane, which is where one would expect the stators to be located. It would be helpful if the authors provided cross-sectional images of the P. mirabilis motor at the level where the stators would be expected to be located. A better way to address this issue, however, would be to determine the in situ structure for the motor of a motA/motB mutant.
Reviewer 2 Report
In their manuscript titled “Structure and assembly of the Proteus mirabilis flagellar motor by cryo-electron tomography” Kaplan et al. report an interesting study on the in situ structure of the flagellar hook basal body in Proteus mirabilis, an important Gram-negative bacterial pathogen. However, some issues need to be addressed. Although the authors prepared Proteus mirabilis cryo-ET samples in both swimming and swarming conditions. Unfortunately, only a single sub-tomogram average was obtained in this manuscript. This limits the ability of the authors to answer the interesting question of how the situ structure of Proteus mirabilis flagellar motor represents the different torque in both swimming and swarming conditions. In addition, the resolution of the sub-tomogram averaged motor structure in this manuscript is lower than the averaged structures that the Jensen group published in EMBO journal 12 years ago (doi: 10.1038/emboj.2011.186). Therefore, the authors need to provide a more detailed discussion of the limitations of the approach they used in current research and the implications of the lower resolution.
Several comments in below.
1, Figure 1, which shows a cartoon figure about E.coli flagellar hook basal body, is neither from the authors' data nor relevant to their current study target, the Proteus mirabilis flagellar motor structure. The authors should consider removing this figure.
2, Figure 2, the main figure, shows a representative tomographic slicer showing fimbriae instead of flagella. This is not appropriate since the research target is the bacterial flagellar motor. It would make more sense for the authors to show flagellar filaments instead of fimbriae.
3, It would be more concise and simple if the authors combined Figure 1-4 with a figure panel. For Figure 4, the image quality needs to be improved to distinguish the different states of flagellar assembly among the A, B, and C.
4, It needs to be clarified what informative message Figure 5 conveys to scientific readers. The authors should provide a more detailed explanation.
5, In lines 174-178, the authors need to confirm that the treatment they used would not disturb the flagellar motor in swarming states, as they stated.
6, In line 186-187, the authors state that they used over 160 electron doses, which would generate strong dose damage. This is concerning, as the cryo-EM field typically uses a total dose below 50 electrons to avoid dose damage issues. The authors need to provide a more detailed explanation and justification for their approach.
Round 2
Reviewer 2 Report
The authors have already made the modification based on the first round of review comments.